# An Adaptive Cooperative Localization Method for Heterogeneous Air-to-Ground Robots Based on Relative Distance Constraints in a Satellite-Denial Environment

**DOI:** 10.3390/s24144543

**Published:** 2024-07-13

**Authors:** Shidong Han, Zhi Xiong, Chenfa Shi

**Affiliations:** 1Navigation Research Center, Nanjing University of Aeronautics and Astronautics, Nanjing 210016, China; han_shidong@nuaa.edu.cn (S.H.); shichenfa@nuaa.edu.cn (C.S.); 2College of Intelligent Engineering, Nanjing Vocational Institute of Railway Technology, Nanjing 210031, China

**Keywords:** cooperative localization, relative distance, relative position, spatial configuration

## Abstract

Cooperative localization (CL) for air-to-ground robots in a satellite-denial environment has become a current research hotspot. The traditional distance-based heterogeneous multiple-robot CL method requires at least four unmanned aerial vehicles (UAVs) with known positions. When the number of known-position UAVs in a cluster collaborative network is insufficient, the traditional distance-based CL method has a certain inapplicability. A novel adaptive CL method for air-to-ground robots based on relative distance constraints is proposed in this paper. Based on a dynamically changing number of known-position UAVs in the cluster collaborative network, the adaptive fusion estimation threshold is set. When the number of known-position UAVs in the cluster cooperative network is large, the real-time dynamic topology characteristics of multiple robots’ spatial geometric configurations are considered. The optimal spatial geometric configuration between UAVs and unmanned ground vehicles (UGVs) is utilized to achieve a high-precision CL solution for UGVs. Otherwise, in the event that the number of known-position UAVs in a cluster collaborative network is insufficient, distance observation constraint information between UAVs and UGVs is retained in real time. Position observation equations for UGVs’ inertial navigation system (INS) have been constructed using inertial-based high-precision relative position constraints and relative distance constraints from historical to current times. The experimental results show that the proposed method achieves adaptive fusion estimation with a dynamically changing number of known-position UAVs in the cluster collaborative network, effectively verifying the effectiveness of the proposed method.

## 1. Introduction

In the past decades, UAVs and UGVs, as highly intelligent agents, have been widely used in the military and civilian fields due to their strong autonomy and flexibility. For example, in the military field, they are used for strategic strikes [1,2], battlefield environment perception and monitoring [3,4], mission rescue [5], and other tasks. In the civilian field, this technology is used for production, express transportation [6], etc. Guided by the intelligent collaborative control strategy, clusters of multiple agents composed of UAVs and UGVs can fully leverage the advantage of resources and effectively improve the success rate of tasks in complex environments. Tasks that cannot be completed by a single agent can be accomplished by way of multiple agents’ cooperation.

High-precision navigation information plays an important role in completing the above tasks in UGVs and UAVs. INS is mounted on the UGVs and UAVs, as an independent dead-reckoning system, because navigation positioning accuracy is difficult to achieve in practical applications due to inertial measurement unit (IMU) error accumulation [7]. Thus, INS is often combined with in-vehicle navigation sensor information to improve navigation positioning performance. When the satellite signals are available, the position and velocity information output from the satellite receiver are employed as observation vectors to form the inertial/satellite loosely integrated navigation system [8]. In addition, the original navigation information obtained from the satellite receiver, such as pseudo-distance and carrier phase, are used to further improve UGVs’ positioning accuracy [9,10]. The actual operating environments of the UGVs are complex and changeable, and the satellite signals are easily disturbed by external environments, which leads to the rejection of satellite signals.

Apart from satellite receivers, internal navigation sensors such as the wheel speed odometer (odo), vision, and radar are widely used in UGVs to provide accurate speed and relative pose constraint information. Many researchers employ the onboard internal navigation sensor information as the system observation constraints to suppress the divergence of INS error and improve the UGV’s positioning performance [11,12]. However, vision- and radar-based relative pose processing algorithms have certain requirements for the operating environment characteristics. When the UGV operates in an area with poor outdoor environment characteristics, the positioning performance of the UGV is difficult to ensure. At the same time, such algorithms have higher requirements for the hardware performance of navigation processors. The navigation system requires a large amount of computation, making it difficult to meet practical application requirements in terms of the real-time performance of navigation algorithms, to a certain extent. In addition, due to the influence of the odometer scale’s factor error, the positioning error of a UGV accumulates and diverges using only the INS/odo loosely integrated navigation system.

In addition to onboard internal navigation sensor information, the massive and rich external relative distance perception information in the cluster collaborative network can serve as observation constraints to improve the positioning accuracy of UGVs in a satellite-denial environment. Under the condition of time synchronization between UAVs and UGVs in the cluster cooperative network, distance perception information can be obtained by using the time-of-arrival (TOA) or time-difference-of-arrival (TDOA) [13,14] methods through a datalink system. Many researchers in the field of cooperative navigation have proposed a series of CL methods to realize multi-robot high-precision CL. For example, combined with onboard magnetic sensor information, Yang proposed a multi-robot cooperative navigation method based on relative distance and magnetic measurement observations [15]. Magnetic sensor information and relative distance information are used as the system observations to construct position observation equations. The positioning error of the dead-reckoning system is effectively suppressed. In addition, Qu Y [16] proposed a CL method for low-cost multi-UAV systems using a relative distance measurement. In the case of at least four UAVs with known positions in the cluster cooperative network, the relative distance information between known-position UAVs and follower UAVs to be located is used as the system observations to construct position observation equations. The principle of spherical intersection is employed to obtain the position closed-form solution of the follower UAV. The positioning performance of the follower UAV in a satellite-denial environment can be effectively improved using the proposed method. In addition, some researchers have applied the thinking logic of the satellite spatial geometry configuration to outdoor cluster UAV formation [17]. By selecting the optimal configuration in real time for CL calculation, the positioning accuracy of the follower UAV in a satellite-denial environment can be improved to some extent.

The above CL methods require at least four UAVs in a cluster cooperative network with known positions. When the number of known-position UAVs in the cluster cooperative network is small and insufficient, traditional distance-based CL methods have a certain inapplicability. In response to the limited number of known-position UAVs in a cluster cooperative network, the airborne magnetic sensor, speed measured sensor, barometric altimeter, and relative distance information are widely employed by researchers to achieve a high-precision CL solution [7].

From the above analysis, we can conclude that current mainstream relative distance-based CL methods are mainly based on the difference in the number of known-position UAVs; thus, the corresponding cooperative localization methods are studied. Considering the number of known-position UAVs in an actual cluster cooperative network is dynamically changed due to being target-driven. Traditional distance-based CL methods based on the specific number of known-position UAVs are somewhat unsuitable. Therefore, this article takes a cluster collaborative network composed of multiple UAVs and multiple UGVs as an example, aimed at the dynamic change in the number of known-position UAVs in a cluster collaborative network, and a novel adaptive CL method for heterogeneous air-to-ground robots based on relative distance constraints is proposed in this paper. According to the dynamic change in the number of known-position UAVs in a cluster collaborative network, the proposed method can be dynamically switched by setting the corresponding fusion threshold. On the basis of constructing a refined error model of UGVs’ INS, when the number of known-position UAVs is large, the optimal spatial geometric configuration is selected in real time using a spatial vector tetrahedron volume minimization mechanism. The relative distance observations between known-position UAVs and UGVs to be located under the optimal spatial geometric configuration are used to realize high-precision CL calculation. On the contrary, when the number of known-position UAVs in a cluster collaborative network is insufficient, the position observation equations of the UGV INS are constructed by saving relative distance observation data between known-position UAVs and UGVs. Utilizing characteristics of high-precision relative position constraints of inertia-based measurements in a short time period, position observation equations of the UGV INS are constructed using relative distance constraints from historical time to current time and inertia-based relative position constraints. Finally, the adaptive extended Kalman filter (AEKF) is designed to perform optimal fusion estimation of the system state vector. The method proposed in this paper can achieve high-precision adaptive CL solution with dynamic change in the number of known-position UAVs in a cluster collaborative network. Thus, the innovations of this paper are summarized as follows:1.Aiming at dynamic change in number of known-position UAVs in a cluster collaborative network driven by tasks, an adaptive CL method for heterogeneous air-to-ground robots based on relative distance constraints is proposed in this paper. The proposed method can achieve adaptive fusion estimation and high-precision CL solution with dynamic change in the number of known-position UAVs.2.In case of an insufficient number of known-position UAVs in a cluster collaborative network, and without the assistance of internal navigation sensor information, the proposed method makes full use of distance constraints from historical times and inertia-based relative position constraints to achieve high-precision CL solution for UGVs.3.By introducing an adaptive fusion threshold, dynamic switching of the proposed method in various CL scenarios can be achieved depending on the number of known-position UAVs in a cluster collaborative network. The system’s adaptive fusion ability is effectively improved using the proposed method.

The rest of the paper is organized as follows. In Section 2, the framework of the proposed adaptive collaborative positioning method is introduced. The proposed adaptive cooperative localization method under various collaborative positioning scenarios is designed in Section 3. In Section 4, the positioning performance of the proposed method against the traditional distance-based CL method and mainstream CL method is compared and analyzed. Finally, the conclusion of this study is summarized in Section 5.

## 2. The Framework of Proposed Adaptive Collaborative Positioning Method

In order to better describe the proposed adaptive collaborative positioning method, the geographic coordinate system is defined as a navigation coordinate system in this paper. In the navigation coordinate system, the *x*-axis points to the east direction, the *y*-axis points to the north direction, the *z*-axis is determined by the right-hand rule. The origin of the navigation coordinate system is located at the centroid of the carrier. The front and right directions in the body coordinate system are pointing to the *x*-axis and y-axis, respectively, and the *z*-axis is determined by the right-hand criterion.

In this paper, a typical target-driven CL scenario for air-to-ground cluster robots based on relative distance constraints in a practical environment is considered, as shown in Figure 1. Indeed, the cluster collaborative network has N heterogeneous robots, of which M are UAV nodes and the rest are UGV nodes. In Figure 1, $(x_{u}^{i},y_{u}^{i},z_{u}^{i}),\;i=1,2,\ldots ,M$ is the location of the UAV, $(x_{G}^{j},y_{G}^{j},z_{G}^{j}),\;j=1,2,\ldots ,N-M$ is the location of the UGV to be located, and $d_{Gj}^{ui}$ is the distance between the UAV node i and UGV node j.

In the above target-driven air-to-ground robot cooperative positioning scenario, the aerial UAVs fly in the available area of satellite signals and are equipped with low-precision Micro-Electro-Mechanical System IMU(MEMS-IMU) sensors, single-point Global Positioning System (GPS) receivers, and datalink communication devices. The UAVs adopt INS/GPS loosely integrated navigation systems to obtain high-precision position information. Due to the influence of external environmental factors such as satellite signal occlusion and multipath effects, satellite signals are denied for UGVs. At the same time, the UGVs are equipped with low-precision MEMS-IMU sensors and datalink communication devices.

We assume that the cluster cooperative network is time-synchronized using the datalink time precision alignment method [18,19]. In the process of CL, UAVs share and transmit their position information to UGVs in real time by datalink system terminals. UGVs receive UAVs’ high-precision position information and measure relative distance perception information with them by the TOA or TDOA method. The UAVs’ high-precision position information and related relative distance constraint information are employed to achieve high-precision CL calculation for UGV. We consider that, in a practical environment, the number of known-position UAVs is influenced by being target-driven, which is dynamically changed over time. In order to realize a high-precision adaptive CL solution for UGVs under dynamic change in the number of known-position UAV nodes, an adaptive cooperative localization method for heterogeneous air-to-ground robots based on relative distance constraints is proposed in this paper. By setting the adaptive adjustment threshold based on the number of known-position UAV nodes, a high-precision adaptive CL solution for UGVs is realized. When the number of known-position UAVs is large in a cluster cooperative network, the spatial geometric configuration advantage between known-position UAVs and UGVs is utilized. The geometric dilution of precision (GDOP) optimization mechanism is used to select the optimal spatial geometric configuration. The UAVs’ position and related distance observation information under the optimal spatial geometric configuration is employed to construct a position observation equation of the UGV INS. Otherwise, in the case that the number of known-position UAVs in the cluster cooperative network is insufficient due to being target-driven, distance observation constraint information between known-position UAVs and UGVs from historical times are maintained in real time. The position observation equations of the UGV INS are constructed using inertia-based high-precision relative position constraints and relative distance constraints from historical times to the current time. The high-precision position closed-form solution of UGV is obtained using the proposed method when the number of known-position UAV nodes is insufficient. The method proposed in this paper has a good adaptive collaborative fusion ability with dynamic change in the number of known-position UAV nodes. The proposed method achieves adaptive CL calculation according to the dynamic change in known-position UAV nodes in the cluster cooperative network. Aiming to locate a UGV node in the cluster cooperative network, the principle schematic diagram of the proposed adaptive cooperative localization method for air-to-ground robots based on relative distance constraints is illustrated in Figure 2.

In Figure 2, $P_{1}^{k},\;P_{1}^{k+1},\;P_{1}^{k+2}$ are the positions of UAV node 1 at time $t_{k}$,$\;t_{k+1}$, and $t_{k+2}$, respectively. $P_{2}^{k},\;P_{2}^{k+1}$ are the positions of UAV node 2 at time $t_{k}$ and time $t_{k+1}$, respectively. $\Delta P_{k}^{k+1},\;\Delta P_{k}^{k+2}$ are the relative position constraints of UGV from time $t_{k}$ to $t_{k+1}$ and $t_{k+2}$, respectively. $d_{k},\;d_{k+1},\;d_{k+2}$ are the relative distance constraints between UAV and UGV at time $t_{k}$, $t_{k+1}$, and $t_{k+2}$, respectively.

## 3. The Design of Proposed Adaptive Cooperative Localization Method under the Various Cooperative Positioning Scenarios

### 3.1. Adaptive Fusion Threshold Setting Based on Number of Known-Position UAVs in Cluster Cooperative Network

In order to enable the method proposed in this paper to achieve adaptive fusion estimation with dynamic change in the number of known-position UAVs in a heterogeneous cluster cooperative network, the dynamic switching of the proposed method is realized by setting adaptive fusion threshold λ based on the number of known-position UAVs in various CL scenarios. Indeed, in the process of CL, the UGV receives communication data packets transmitted by UAVs in real time. The ID numbers and corresponding position information of UAVs be obtained by UGV through the packet protocol analysis. According to the ID number of the UAV, the vehicle navigation computer can automatically obtain in real time the number of known-position UAVs in the cluster cooperative network. In the case that the number of known-position UAVs in the cluster cooperative network is larger than three, the adaptive fusion threshold of the proposed method is set to $\lambda_{1}$. In addition, when there are only two UAVs with known positions in the dynamic cooperative network, the UGV INS lacks a sufficient number of distance observation constraints to construct position observation equations. Combined with distance observation constraints between known-position UAVs and UGVs from historical times, system position observation equations are constructed using inertia-based high-precision relative position constraints. The adaptive fusion threshold in this CL scenario is set to $\lambda_{2}$. Otherwise, the cluster cooperative network has only one UAV with known position, and the adaptive fusion threshold of the proposed method is set to $\lambda_{3}$. By setting the adaptive fusion threshold, the method proposed in this paper achieves dynamic switching and adaptive fusion estimation in various CL scenarios. Under the condition that the measured accuracy of the relative distance between UAVs and UGV is consistent, state estimation accuracy and positioning performance of the UGV in a satellite-denial environment can be further improved using the proposed method.

### 3.2. State Propagation Model

Based on the error characteristic of the UGV’s inertial sensor in a complex environment [20], the state recursive equation of the UGV INS is constructed in this paper. The state propagation model of the proposed adaptive CL method can be constructed as follows:(1)X(k+1)=f(X(k),k)+w(k)
where
(2)$$\begin{array}{ll} X(k)= & [\varphi_{E}(k),\varphi_{N}(k),\varphi_{U}(k),\delta v_{E}(k),\delta v_{N}(k),\delta v_{U}(k),\delta L(k),\\ & \delta \lambda (k),\delta h(k),\varepsilon_{bx}(k),\varepsilon_{by}(k),\varepsilon_{bz}(k),\varepsilon_{rx}(k),\varepsilon_{ry}(k),\\ & \varepsilon_{rz}(k),\nabla_{ax}(k),\nabla_{ay}(k),\nabla_{az}(k){]}^{T} \end{array}$$

X(k) is the system state vector, which can be, respectively, expressed as follows: platform misalignment angle error of UGV INS, velocity error of UGV INS, position error of UGV INS, gyroscope constant bias, gyroscope first-order Markov white noise, and accelerometer first-order Markov white noise. w(k) is the system noise input matrix. We assume that w(k) is Gaussian white noise with mean value of zero and satisfies $E\{w(k)w^{T}(k)\}=Q(k)$. The state transition Jacobian matrix F(k+1,k) and system noise control input Jacobian matrix W(k+1,k) can be, respectively, expressed as:(3)$$\left\{\begin{array}{l} F(k+1,k)=\frac{\partial f(X(k),k)}{\partial X}{|}_{X=\hat{X}(k|k)}\\ W(k+1,k)=\frac{\partial f(X(k),k)}{\partial u}{|}_{X=\hat{X}(k|k)} \end{array}\right.$$

### 3.3. Position Observation Equation of UGV INS with a Large Number of Known-Position UAVs

Under the condition of time synchronization of the cluster cooperative network, the spherical intersection principle can be used to achieve high-precision CL calculation using the position information of three UAVs and related distance observation information [14]. Driven by UAV tasks, when the number of known-position UAVs in the cluster cooperative network is larger than three, real-time dynamic topology change in the spatial geometric configuration between UAVs and UGV is considered. The optimal spatial geometric configuration is optimized using a GDOP optimization mechanism. The UAVs’ positions and related distance observation information in the optimal spatial geometric configuration are employed to realize the high-precision CL solution, which can effectively improve the UGV’s positioning performance.

We assume that the three-dimensional position information of UAVs in Figure 1 is $\left(x_{1},y_{1},z_{1}\right)$, $\left(x_{2},y_{2},z_{2}\right)$, $\left(x_{3},y_{3},z_{3}\right)$, and $(x_{i},y_{i},z_{i}),\;i\geq 3$, respectively, obtained by the INS/GPS loosely integrated navigation system. The position information of the UGV to be located in Figure 1 is $\left(x_{j},y_{j},z_{j}\right)$, obtained by inertial integration operation. The UGV on the ground obtains relative distance observation information with known-position UAVs in real time by the TDOA method. The relative distance observation constraints between known-position UAVs and UGV can be reported as follows:(4)$$\left\{\begin{array}{l} {\left(x_{1}-x_{j}\right)}^{2}+{\left(y_{1}-y_{j}\right)}^{2}+{\left(z_{1}-z_{j}\right)}^{2}=d_{1j}^{2}\\ {\left(x_{2}-x_{j}\right)}^{2}+{\left(y_{2}-y_{j}\right)}^{2}+{\left(z_{2}-z_{j}\right)}^{2}=d_{2j}^{2}\\ \cdots \\ {\left(x_{i}-x_{j}\right)}^{2}+{\left(y_{i}-y_{j}\right)}^{2}+{\left(z_{i}-z_{j}\right)}^{2}=d_{ij}^{2} \end{array}\right.$$
where $d_{1j}^{2},d_{2j}^{2},\ldots d_{ij}^{2}$ represent relative distance observation constraints between known-position UAVs and the UGV to be located. The relative distance observation constraints among robots are non-linear equations with strong nonlinearity. Thus, the linearization Taylor expansion of Equation (4) is carried out, and Equation (4) can be rewritten as:(5)$$f_{i}(X)=f_{i}(\hat{X})+\frac{\partial f(X)}{\partial X}{|}_{X=\hat{X}}(X-\hat{X})+H.O.T$$
where X is the system state vector composed of the error of the UGV INS. $\hat{X}$ is the state estimated value. $f_{i}(X)$ is the relative distance observation function related to the UGV’s state. ***H.O.T*** is the high-order term above the second order of linearization Taylor expansion, which is not considered in this paper. In order to better describe the spatial configuration optimization strategy proposed in this paper, Equation (5) can be rewritten to the following form:(6)δΩ=HδX

In Equation (6), $\delta \Omega =f(X)-f(\hat{X})$ is the distance observation residual, and $H=\frac{\partial f(X)}{\partial X}{|}_{X=\hat{X}}$ is the observation Jacobian matrix associated with relative distance observation constraints. $\delta X=X-\hat{X}$ is the state estimation error. Considering the non-singularity of the Jacobian matrix of the relative distance observations between the known-position UAVs and UGV, state estimation error in Equation (6) can be expressed in the following form:(7)$$\delta X=H^{-1}\delta \Omega$$
where $H^{-1}$ represents the inverse matrix of relative distance observation Jacobian H. Thus, the Root Mean Square Error (RMSE) matrix of the state estimation error is shown in Equation (8).
(8)$$\begin{array}{ll} E\left[\delta X{\left(\delta X\right)}^{T}\right] & =E\left[H^{-1}\delta \Omega {\left(\delta \Omega \right)}^{T}{\left(H^{-1}\right)}^{T}\right]\\ & =H^{-1}E\left[\delta \Omega {\left(\delta \Omega \right)}^{T}\right]{\left(H^{T}\right)}^{-1}=\sigma {\left(H^{T}H\right)}^{-1} \end{array}$$

The trace of the RMSE matrix is employed in this paper as the real time optimization strategy for known position UAVs. In general, the trace of RMSE matrix is commonly represented as GDOP. The spatial geometric configuration with the minimum GDOP value is effectively selected as the optimal spatial geometric configuration in this paper. UAVs with known positions and corresponding distance observation information in the optimal spatial geometric configuration are used to realize the high-precision CL calculation of UGV. Thus, the real-time optimization strategy based on the spatial geometric configuration proposed in this paper is reported in Equation (9).
(9)$$\min\left(\mathrm{trace}\left(\sigma {\left(H^{T}H\right)}^{-1}\right)\right)$$

Using the configuration optimization strategy mentioned above, the known-position UAVs and related distance observation information under the optimal spatial geometric configuration are employed to construct a position observation equation of the UGV INS, which can further improve state estimation accuracy and positioning accuracy of the UGV in the satellite signal-denial area. We assume that relative distance measurement noises between known-position UAVs and UGV in the dynamic cooperative network are mutually independent, and satisfy Gaussian-type white noise with the mean value of zero, and covariance of a certain value $R_{\mathrm{distance}}$. In the case of a large number of known-position UAVs in a dynamic cooperative network, the position observation equation of the UGV INS is constructed by first-order linearization Taylor expansion of Equation (4).
(10)$$\begin{array}{l} Z_{k}^{\mathrm{distance}}=H_{k}C_{e}^{n}X_{k}+V_{k}\\ H_{k}=\left[\begin{array}{ccc} \frac{\partial f_{1}}{\partial x} & \frac{\partial f_{1}}{\partial y} & \frac{\partial f_{1}}{\partial z}\\ \frac{\partial f_{2}}{\partial x} & \frac{\partial f_{2}}{\partial y} & \frac{\partial f_{2}}{\partial z}\\ \frac{\partial f_{3}}{\partial x} & \frac{\partial f_{3}}{\partial y} & \frac{\partial f_{3}}{\partial z} \end{array}\right] \end{array}$$
where $Z_{k}^{\mathrm{distance}}$ is the system observation vector formed by the difference between the relative distance calculated value and actual measured value. $f=\left[f_{1},f_{2},f_{3}\right]$ illustrate the relative distance measurement function between known-position UAVs and UGV in the optimal spatial geometric configuration. $C_{e}^{n}$ is the attitude transition matrix from the earth coordinate system to the navigation coordinate system. $H_{k}$ is the observation Jacobian matrix corresponding to distance observation constraints under the optimal spatial geometric configuration. $V_{k}$ is the relative distance measurement noise matrix.

### 3.4. Position Observation Equation of UGV INS with Two Known-Position UAVs in Cluster Cooperative Network

Under the control of the UAV target-driven strategy, the number of known-position UAVs in the cluster cooperative network is dynamically changed in real time. When the number of known-position UAVs is insufficient, and there are only two UAVs with known positions in the cluster cooperative network, the UGV on the ground cannot obtain a sufficient number of distance observations to construct the position observation equation of the UGV INS. The positioning error of the UGV to be located accumulates and diverges for a long time using the traditional distance-based CL method. In order to improve the UGV’s state estimation accuracy and positioning accuracy in the case that the number of known-position UAVs is insufficient, distance observation constraints between UAVs and UGV from the historical time are saved in this paper. The position observation equation of the UGV INS is constructed by using inertia-based short-term high-precision relative position constraint and relative distance constraints at adjacent times. The spatial position relationship between the UGV and two known-position UAVs at adjacent times is reported in Figure 3. In Figure 3, $\left(x_{u1}^{k},y_{u1}^{k},z_{u1}^{k}\right)$, $\left(x_{u1}^{k+1},y_{u1}^{k+1},z_{u1}^{k+1}\right)$ represent the position information of UAV node 1 at times $t_{k}$ and $t_{k+1}$, respectively, obtained by INS/GPS loosely integrated navigation systems. $\left(x_{u2}^{k},y_{u2}^{k},z_{u2}^{k}\right)$ and $\left(x_{u2}^{k+1},y_{u2}^{k+1},z_{u2}^{k+1}\right)$ are the position information of UAV node 2 at times $t_{k}$ and $t_{k+1}$, respectively, obtained by INS/GPS loosely integrated navigation systems. $d_{u1,G}^{k},\;d_{u1,G}^{k+1},\;d_{u2,G}^{k},\;d_{u2,G}^{k+1}$ are the relative distance observation constraints between known-position UAVs and UGV at times $t_{k}$ and $t_{k+1}$, respectively. At the same time, $\left(x_{G}^{k},y_{G}^{k},z_{G}^{k}),(x_{G}^{k+1},y_{G}^{k+1},z_{G}^{k+1}\right)$ are the position information of the UGV to be located at times $t_{k}$ and $t_{k+1}$, respectively.

In the case of an insufficient number of known-position UAVs in the dynamic cooperative network, inertia-based relative position constraints play an important role in constructing the position observation equation of the UGV INS. When the initial position and variance of the UGV are accurately known, the inertial sensor has a higher relative position estimation accuracy in a short time period according to the inertial arrangement mechanism. The inertial sensor is generally composed of a three-axis gyroscope and a three-axis accelerometer, which is used for sensitive angular velocity and linear acceleration information of the UGV. Affected by constant bias of the inertial sensor, the actual measured value of the inertial sensor be expressed as:(11)$$\left\{\begin{array}{l} {\tilde{\omega }}_{ib}^{b}={\bar{\omega }}_{ib}^{b}+b_{g}+\eta_{g}\\ {\tilde{a}}_{ib}^{b}={\bar{a}}_{ib}^{b}+b_{a}+\eta_{a} \end{array}\right.$$
where ${\tilde{\omega }}_{ib}^{b}$ and ${\tilde{a}}_{ib}^{b}$ are the actual measured value of the three-axis gyroscope and accelerometer, respectively. ${\bar{\omega }}_{ib}^{b}$ and ${\bar{a}}_{ib}^{b}$ are the true measurement value of the three-axis gyroscope and accelerometer, respectively. $b_{g}$ and $b_{a}$ are the constant bias of the three-axis gyroscope and accelerometer, respectively. $\eta_{g}$ and $\eta_{a}$ are the Gaussian measurement white noise of the three-axis gyroscope and accelerometer. Under the condition that the initial state of the UGV is accurately known, the next state of the UGV is predicted using Equation (12).
(12)$$\begin{array}{l} q_{k+1}=q_{k}+0.5q_{k}{}^{\circ}\omega_{nb}^{b}T\\ v_{k+1}^{n}=v_{k}^{n}+(2\omega_{ie}^{n}+\omega_{en}^{n})v_{k}^{n}+\left(f_{k}^{n}+g^{n}\right)T\\ L_{k+1}=L_{k}+\frac{v_{k}^{N}T}{\left(R_{M}+h_{k}\right)}\\ \lambda_{k+1}=\lambda_{k}+\frac{v_{k}^{E}T}{(R_{N}+h_{k})\cos L_{k}}\\ h_{k+1}=h_{k}+v_{k}^{U}T \end{array}$$
where $q_{k},\;q_{k+1},\;v_{k}^{n},\;v_{k+1}^{n}$ are the attitude quaternion and velocity information of the UGV at times $t_{k}$ and $t_{k+1}$, respectively. $\left(\lambda_{k},L_{k},h_{k}\right)$ and $\left(\lambda_{k+1},L_{k+1},h_{k+1}\right)$ are the position information of the UGV at times $t_{k}$ and $t_{k+1}$, respectively. $\omega_{nb}^{b}$ is the projection of the angular velocity of the body coordinate system relative to the geographic coordinate system in the body coordinate system. $\omega_{ie}^{n}$ is the projection of the angular velocity of the earth coordinate system relative to the inertial coordinate system in the geographic coordinate system. $\omega_{en}^{n}$ is the projection of the angular velocity of the geographic coordinate system relative to the earth coordinate system in the geographic coordinate system. $R_{M}$ and $R_{N}$ are the curvature radius of the earth’s meridional and unitary circles, respectively. $g^{n}$ is the earth’s gravity vector in the geographic coordinate system. T is the time interval period at adjacent sampling of the UGV INS.

With the rapid development of modern intelligent sensor technology, the sampling frequency of the inertial sensor is generally higher than the relative distance measurement between the UAV and UGV. Indeed, the sampling frequency of the inertial sensor in practical applications is generally 200 Hz, while the sampling frequency of the relative distance measurement is 10 Hz. The time sequence schematic diagram of the inertial sensor and relative distance observation between the UAV and UGV is illustrated in Figure 4.

From Figure 4, we can conclude that in the adjacent sampling interval of the relative distance observation, the UGV’s state requires multiple inertial integration calculations. At the adjacent sampling interval of the relative distance observation constraint, the inertia-based relative position constraint equation of the UGV from time $t_{k}$ to time $t_{k+1}$ can be derived and constructed as follows.
(13)$$\begin{array}{l} \Delta {\tilde{R}}_{t_{k}}^{t_{k+1}}=\prod_{k=t_{k}}^{t_{k+1}-1}\exp[({\tilde{\omega }}_{ibk}^{b}-b_{g}^{k}-\eta_{g}^{k})T]\\ \Delta {\tilde{v}}_{t_{k}}^{t_{k+1}}=\sum_{k=t_{k}}^{t_{k+1}-1}R_{t_{k}}^{T}R_{k}({\tilde{a}}_{ibk}^{b}-b_{a}^{k}-\eta_{a}^{k})T\\ \Delta {\tilde{p}}_{t_{k}}^{t_{k+1}}=\sum_{k=t_{k}}^{t_{k+1}-1}\left[R_{t_{k}}^{T}(v_{k}-v_{t_{k}}-gT)T+\frac{1}{2}R_{t_{k}}^{T}R_{k}({\tilde{a}}_{ibk}^{b}-b_{a}^{k}-\eta_{a}^{k})T^{2}\right] \end{array}$$
where $\Delta {\tilde{R}}_{t_{k}}^{t_{k+1}},\;\Delta {\tilde{v}}_{t_{k}}^{t_{k+1}},\;\Delta {\tilde{p}}_{t_{k}}^{t_{k+1}}$ are the relative attitude constraint, relative velocity constraint, and relative position constraint of the UGV from time $t_{k}$ to time $t_{k+1}$.

Combined with the relative distance observation constraints between UAVs and UGV at adjacent times and the relative position constraint of the UGV from time $t_{k}$ to time $t_{k+1}$, the position observation equation of the UGV INS at time $t_{k+1}$ can be derived and constructed, as shown in Equation (14).
(14)$$\left\{\begin{array}{l} {\left(x_{u1}^{k+1}-x_{G}^{k+1}\right)}^{2}+{\left(y_{u1}^{k+1}-y_{G}^{k+1}\right)}^{2}+{\left(z_{u1}^{k+1}-z_{G}^{k+1}\right)}^{2}={\left(d_{u1,G}^{k+1}\right)}^{2}\\ {\left(x_{u2}^{k+1}-x_{G}^{k+1}\right)}^{2}+{\left(y_{u2}^{k+1}-y_{G}^{k+1}\right)}^{2}+{\left(z_{u2}^{k+1}-z_{G}^{k+1}\right)}^{2}={\left(d_{u2,G}^{k+1}\right)}^{2}\\ {\left(x_{u1}^{k}-x_{G}^{k+1}+\Delta {\tilde{p}}_{x}\right)}^{2}+{\left(y_{u1}^{k}-y_{G}^{k+1}+\Delta {\tilde{p}}_{y}\right)}^{2}+{\left(z_{u1}^{k}-z_{G}^{k+1}+\Delta {\tilde{p}}_{z}\right)}^{2}={\left(d_{u1,G}^{k}\right)}^{2}\\ {\left(x_{u2}^{k}-x_{G}^{k+1}+\Delta {\tilde{p}}_{x}\right)}^{2}+{\left(y_{u2}^{k}-y_{G}^{k+1}+\Delta {\tilde{p}}_{y}\right)}^{2}+{\left(z_{u2}^{k}-z_{G}^{k+1}+\Delta {\tilde{p}}_{z}\right)}^{2}={\left(d_{u2,G}^{k}\right)}^{2} \end{array}\right.$$
where $\Delta {\tilde{p}}_{t_{k}}^{t_{k+1}}=\left[\begin{array}{ccc} \Delta {\tilde{p}}_{x} & \Delta {\tilde{p}}_{y} & \Delta {\tilde{p}}_{z} \end{array}\right]$ denotes the three-axis relative position constraint of the UGV from time $t_{k}$ to time $t_{k+1}$. Considering that the distance observation constraints between UAVs and UGV are nonlinear observations with strong nonlinearity, the first-order linearization Taylor expansion of Equation (14) is made, ignoring the higher-order terms above the second order. Under the condition that there are only two known-position UAVs in the dynamic cluster cooperative network, the position observation equation of the UGV INS at time $t_{k+1}$ can be constructed as follows:(15)$$\begin{array}{c} Z_{k+1}^{P}=H_{k+1}^{P}X_{k+1}+v_{p}\\ H_{k+1}^{P}={\left[\begin{array}{ccc} 0_{4\times 6} & H_{4\times 3}^{\rho } & 0_{4\times 6} \end{array}\right]}_{4\times 18}\\ H_{4\times 3}^{p}=\left[\begin{array}{ccc} -(R_{N}+h)[f_{11}\sin L\cos\lambda +f_{12}\sin L\sin\lambda ] & \left(R_{N}+h)[f_{12}\cos L\cos\lambda -f_{11}\cos L\sin\lambda \right] & f_{11}\cos L\cos\lambda +f_{12}\cos L\sin\lambda +f_{13}\sin L\\ -(R_{N}+h)[f_{21}\sin L\cos\lambda +f_{22}\sin L\sin\lambda ] & \left(R_{N}+h)[f_{22}\cos L\cos\lambda -f_{21}\cos L\sin\lambda \right] & f_{21}\cos L\cos\lambda +f_{22}\cos L\sin\lambda +f_{23}\sin L\\ -(R_{N}+h)[f_{31}\sin L\cos\lambda +f_{32}\sin L\sin\lambda ] & \left(R_{N}+h)[f_{32}\cos L\cos\lambda -f_{31}\cos L\sin\lambda \right] & f_{31}\cos L\cos\lambda +f_{32}\cos L\sin\lambda +f_{33}\sin L\\ -(R_{N}+h)[f_{41}\sin L\cos\lambda +f_{42}\sin L\sin\lambda ] & \left(R_{N}+h)[f_{42}\cos L\cos\lambda -f_{41}\cos L\sin\lambda \right] & f_{41}\cos L\cos\lambda +f_{42}\cos L\sin\lambda +f_{43}\sin L \end{array}\right] \end{array}$$
where $Z_{k+1}^{P}$ denotes the system position observation vector. $H_{k+1}^{P}$ denotes the position observation Jacobian matrix corresponding to the distance observation equation. $X_{k+1}$ is the system state vector. $v_{p}$ represents the relative distance measurement noise matrix. $f_{i1}=\frac{\partial f_{i}}{\partial x}$, $f_{i2}=\frac{\partial f_{i}}{\partial y}$, $f_{i3}=\frac{\partial f_{i}}{\partial z},\;i=1,2,3,4$ are the directional cosines of the position observation equation in three directions, respectively.

### 3.5. Position Observation Equation of UGV INS with Only Single Known-Position UAV in Cluster Cooperative Network

In the case of a dynamic cooperative network with only one known-position UAV node, the single relative distance observation information is received by the UGV in real time. Without the assistance of onboard internal navigation sensor information, the UGV is located on a spherical surface with the UAV’s position as the center and relative distance as the radius. Using the single relative distance information as the observation constraint of the UAV INS, the position closed-form solution of the UGV cannot be obtained in real time. The positioning error of the UGV accumulates and diverges over time using the traditional distance-based CL method. In order to improve the UGV’s positioning accuracy under the condition of insufficient distance observation constraints, the distance observation information between the UAV and UGV from historical times is preserved. The position observation equation of the UGV INS at time $t_{k+2}$ is constructed by using historical distance observation constraints and inertia-based short-term high-precision relative position constraints.

We aim to construct a dynamic cluster cooperative network composed of a known-position UAV and a UGV to be located. The spatial position relationship between the UGV and single UAV at three consecutive relative distance sampling periods is presented in Figure 5.

In Figure 5, $\left(d_{uG}^{k},d_{uG}^{k+1},d_{uG}^{k+2}\right)$ represent the relative distance observation constraints between UAV and UGV at times $t_{k}$, $t_{k+1}$, and $t_{k+2}$, respectively. $\left(x_{u}^{k},y_{u}^{k},z_{u}^{k}\right)$, $\left(x_{u}^{k+1},y_{u}^{k+1},z_{u}^{k+1}\right)$, $\left(x_{u}^{k+2},y_{u}^{k+2},z_{u}^{k+2}\right)$ are the position information of the UAV at times $t_{k}$, $t_{k+1}$, and $t_{k+2}$, respectively, obtained by the INS/GPS loosely integrated navigation system. $(x_{G}^{k},y_{G}^{k},z_{G}^{k}),\;(x_{G}^{k+1},y_{G}^{k+1},z_{G}^{k+1}),$ $\left(x_{G}^{k+2},y_{G}^{k+2},z_{G}^{k+2}\right)$ are the position information of the UGV at times $t_{k}$, $t_{k+1}$, and $t_{k+2}$, obtained by inertial integration operation.

We assume that the system time between the UAV and UGV is synchronized in the dynamic cluster cooperative network. By retaining distance observation constraints between the UAV and UGV for three consecutive sampling periods, we can conclude from Figure 5 that the position closed-form solution of the UGV at time $t_{k+2}$ is obtained using the principle of multi-spherical intersection. Using high-precision position information of the UAV from $t_{k}$ to $t_{k+1}$, $t_{k+2}$, the distance observation constraint information between the UAV and UGV at various sampling periods, and inertia-based relative position constraint information from time $t_{k}$ to $t_{k+1}$ and from time $t_{k}$ to $t_{k+2}$, the position observation equation of the UGV INS at time $t_{k+2}$ is constructed. Thus, the position observation equation of the UGV INS at time $t_{k+2}$ can be derived and constructed as follows:(16)$$\left\{\begin{array}{l} {\left(x_{u}^{k}-x_{G}^{k}\right)}^{2}+{\left(y_{u}^{k}-y_{G}^{k}\right)}^{2}+{\left(z_{u}^{k}-z_{G}^{k}\right)}^{2}={\left(d_{uG}^{k}\right)}^{2}\\ {\left(x_{u}^{k+1}-x_{G}^{k+1}\right)}^{2}+{\left(y_{u}^{k+1}-y_{G}^{k+1}\right)}^{2}+{\left(z_{u}^{k+1}-z_{G}^{k+1}\right)}^{2}={\left(d_{uG}^{k+1}\right)}^{2}\\ {\left(x_{u}^{k+2}-x_{G}^{k+2}\right)}^{2}+{\left(y_{u}^{k+2}-y_{G}^{k+2}\right)}^{2}+{\left(z_{u}^{k+2}-z_{G}^{k+2}\right)}^{2}={\left(d_{uG}^{k+2}\right)}^{2} \end{array}\right.$$
where
(17)$$\begin{array}{l} x_{G}^{k+1}=x_{G}^{k}+\Delta x_{k,k+1}\;x_{G}^{k+2}=x_{G}^{k}+\Delta x_{k,k+2}\\ y_{G}^{k+1}=y_{G}^{k}+\Delta y_{k,k+1}\;y_{G}^{k+2}=y_{G}^{k}+\Delta y_{k,k+2}\\ z_{G}^{k+1}=z_{G}^{k}+\Delta z_{k,k+1}\;z_{G}^{k+2}=z_{G}^{k}+\Delta z_{k,k+2} \end{array}$$

In Equation (17), $\left(\Delta x_{k,k+1},\Delta y_{k,k+1},\Delta z_{k,k+1}\right)$ represent the inertia-based relative position constraints of the UGV from time $t_{k}$ to time $t_{k+1}$, obtained by Equation (13). Meanwhile, $\left(\Delta x_{k,k+2},\Delta y_{k,k+2},\Delta z_{k,k+2}\right)$ represent the inertia-based relative position constraints of the UGV from time $t_{k}$ to time $t_{k+2}$, also obtained by Equation (13). It can be seen from Equation (16) that in the case of an insufficient number of UAV nodes and related distance observation constraints in the dynamic cooperative network, inertia-based relative position constraints play an important role in constructing the position observation equation of the UGV INS at time $t_{k+2}$. When the initial state $X_{0}$ and variance $P_{0}$ are accurately known in advance, short-term relative position constraints based on an inertial recurrence mechanism have higher estimation accuracy. Under the condition of measurement accuracy and sampling frequency of the UGV’s onboard inertial navigation sensor, the higher the sampling frequency of the relative distance observation, the higher the estimation accuracy of the relative distance constraint for cooperative positioning calculation. By fully utilizing the characteristic of inertia-based short-term high-precision relative position constraints, the position closed-form solution of the UGV at time $t_{k+2}$ is obtained using the proposed method. Without the assistance of the onboard endogenous navigation sensor information, the positioning accuracy and cooperative positioning performance of the UGV in a satellite-denial environment can be effectively improved using the proposed method.

The first-order linearization Taylor expansion of Equation (16) is made to construct the position observation equation of the UGV INS at time $t_{k+2}$, which can be expressed as:(18)$$Z_{p}(k+2)=H_{p}(k+2)\delta P(k+2)+V(k+2)$$
where $Z_{p}(k+2)$ is the position observation vector. $H_{p}(k+2)$ is the position observation Jacobian matrix. δP(k+2) is the position error of the UGV in the earth coordinate system. V(k+2) is the measurement noise matrix related to the relative distance observations.
(19)$$H_{p}(k+2)={\left[\begin{array}{ccccc} 0_{1\times 6} & \frac{\partial f_{k}}{\partial x} & \frac{\partial f_{k}}{\partial y} & \frac{\partial f_{k}}{\partial z} & 0_{1\times 6}\\ 0_{1\times 6} & \frac{\partial f_{k+1}}{\partial x} & \frac{\partial f_{k+1}}{\partial y} & \frac{\partial f_{k+1}}{\partial z} & 0_{1\times 6}\\ 0_{1\times 6} & \frac{\partial f_{k+2}}{\partial x} & \frac{\partial f_{k+2}}{\partial y} & \frac{\partial f_{k+2}}{\partial z} & 0_{1\times 6} \end{array}\right]}_{3\times 18}$$
where $\left[f_{k},f_{k+1},f_{k+2}\right]$ are the position observation equations in Equation (16). Considering that the UGV’s position error in the system state vector is located at the geographic coordinate system, according to the spatial attitude transformation relationship between the geographic coordinate system and the earth coordinate system, the position observation equation of the UGV INS at time $t_{k+2}$ can be rewritten as:(20)$$Z_{p}(k+2)=H_{p}(k+2)H_{T}(k+2)X(k+2)+V(k+2)$$
where $H_{T}(k+2)$ is the attitude transformation matrix between the geographic coordinate system and earth coordinate system, which can be expressed as:(21)$$H_{T}(k+2)=\left[\begin{array}{ccc} -(R_{N}+h)\sin L\cos\lambda & -(R_{N}+h)\cos L\sin\lambda & \cos L\cos\lambda \\ -(R_{N}+h)\sin L\sin\lambda & (R_{N}+h)\cos L\cos\lambda & \cos L\sin\lambda \\ {}[R_{N}{\left(1-f\right)}^{2}+h]\cos L & 0 & \sin L \end{array}\right]$$

On the basis of constructing the above-mentioned system observation equations in various CL scenarios, the optimal estimation of system state variables is another key technology to improve the UGV’s positioning performance. From the perspective of practical application, system real-time performance and computational complexity are comprehensively considered in this paper, and the AEKF is designed to achieve optimal estimation and online compensation of state variables. For details about AEKF, the reader is referred to [21,22]. The pseudocode of the proposed adaptive cooperative localization method for heterogeneous air-to-ground robots based on relative distance constraints is illustrated in Algorithm 1.
**Algorithm 1.** The pseudocode of proposed adaptive cooperative localization method for heterogeneous air-to-ground robots based on relative distance constraints1.  Initialization     Cooperative network time synchronization and state initialization X0,P02.  Set adaptive fusion threshold λ based on dynamic change in number of known-position UAVs in dynamic cluster cooperative network3.  State recursive equation of UGV based on Equation (1)4.  Construct position observation equation of UGV INS based on the different number of known-position UAVs in cluster cooperative network     If (λ=λ1)     Construct position observation equation of UGV INS at time tk based on Equation (10)     Else if (λ=λ2)     Construct position observation equation of UGV INS at time tk+1 based on Equation (15)     Else if (λ=λ3)     Construct position observation equation of UGV INS at time tk+2 based on Equation (20)5.  State optimal fusion estimation based on AEKF6.  Optimal state estimation value of UGV

## 4. Multiple Heterogeneous Robot Cooperative Localization Test and Results Analysis

In order to verify the effectiveness of the proposed method under a dynamic change in the number of known-position UAV nodes in a cluster cooperative network, a cluster cooperative network composed of five UAVs and a UGV to be located is used as an example to verify the cooperative positioning performance of the proposed method on a campus playground. Before the experiment, time synchronization between the UAVs and UGV is completed using GPS system time. The initial position of the UGV is calibrated using a single-point GPS receiver. In the experiment, the UAVs and UGV operated autonomously at different altitudes according to preset waypoints. Each UAV is equipped with a low-precision MTI-G-710 MEMS-IMU, a single-point Novatel GPS receiver, and an Ultra-Wideband (UWB) navigation sensor. The position information of the UAV is gained by the INS/GPS loosely integrated navigation system. The low-precision FSS-IMU6132 MEMS-IMU, differential mode Real-time Kinematic (RTK) satellite receiver, and UWB navigation sensors are mounted on the UGV. In addition, in order to better compare and analyze the CL performance of the proposed method against the current mainstream CL method, the UGV is additionally equipped with a wheel speed odometer and a barometric altimeter sensor. A UWB navigation sensor is also used in the experiment to measure the relative distance perception information between the UGV and UAV. The sampling frequencies of the UAV’s MEMS-IMU and UGV’s MEMS-IMU are set to 100 Hz. The sampling frequencies of the wheel speed odometer, barometric altimeter, and UWB navigation sensors are set to 50 Hz, respectively, while the sampling frequency of the GPS satellite receiver is set to 1 Hz. The performance parameters of the heterogeneous navigation sensors mounted on the UGV in the experiment are shown in Table 1. The various navigation sensors mounted on the UAV and UGV are illustrated in Figure 6.

During the experiment, the adaptive collaborative fusion thresholds are set to $\lambda_{1}=3$, $\lambda_{2}=2$, and $\lambda_{3}=1$, respectively, according to number of known-position UAV nodes in the dynamic cluster cooperative network. Before the experiment, the five UAV nodes have the same attitude and velocity information. The initial attitude, initial velocity, and initial position information of the UAV and UGV in the experiment is illustrated in Table 2.

It should be noted that position information output from the RTK satellite receiver is only used as a reference for cooperative positioning performance evaluation, and does not participate in CL calculation. During the experiment, relative distance observation information between the UAV and UGV is measured in real time through the UWB navigation sensor. The heterogeneous navigation sensor data are stored in the navigation computer processor, and the cooperative positioning performance of the proposed method is verified and analyzed by data post-processing. The CL scenario between UAV and UGV nodes based on relative distance constraints on the outdoor campus playground is reported in Figure 7. Three-dimensional autonomous motion trajectories of UAV and UGV nodes are presented in Figure 8. The total CL time is about 210 s.

For the purpose of better validating the adaptive fusion ability of the proposed adaptive CL method under a dynamic change in the number of UAV nodes in the cluster cooperative network, the various cooperative localization scenarios are simulated by utilizing distance observation constraint information with different numbers of UAV nodes.

### 4.1. Experimental Verification and Analysis with Multiple UAV Nodes

When there is a larger number of known-position UAVs in the dynamic cluster cooperative network, the distance observation information received by the UGV is the largest. The spatial geometric configuration between the UAVs and UGV is one of the key factors affecting the positioning performance of the UGV. In order to improve the UGV’s positioning accuracy in a satellite-denial environment, the optimal spatial geometric configuration between the known-position UAVs and UGV to be located is taken into account in this paper. The corresponding distance observation constraint information under the optimal spatial geometric configuration are employed to construct the position observation equation of the UGV INS, which can effectively improve the UGV’s positioning performance. Figure 9 illustrates the GDOP value between UGV and UAV nodes in the experiment.

In order to verify the cooperative positioning performance of the proposed configuration optimization CL method, the RMSE of the UGV’s positioning error is used as an evaluation reference [19]. Table 3 gives the RMSE of the UGV’s positioning error using distance observation information in various spatial geometric configurations.

In Table 3, GDOP123 represents the GDOP value between the UGV and UAV node 1, UAV node 2, and UAV node 3. The definition of other GDOPs is consistent with GDOP123.

It can be seen from Table 3 that the configuration optimization CL method proposed in this paper can fully utilize the configuration advantage between the UGV and UAVs to perform a cooperative positioning solution. Under certain conditions of UAV positioning accuracy and relative distance measurement accuracy of the UWB navigation sensor, we use distance observation information under the optimal spatial geometry configuration to construct the system observation equation, which can improve the UGV’s positioning accuracy in a satellite-denial environment to a certain extent.

At present, the mainstream distance-based CL method mainly utilizes internal sensor information and external relative distance perception information as system observation, to construct system observation equations [7]. In this paper, the wheel speed odometer information, barometric altimeter information, and distance perception information with UAV nodes are used as system observation constraints. The velocity and position observation equations of the UGV INS are constructed using above navigation sources to verify the performance of the proposed method.

For the purpose of better visually comparing and analyzing the performance of the proposed configuration optimization CL method from different perspectives, taking the spatial geometry configuration between UAV nodes 1, 2, 3 and the UGV as an example, the positioning error curves of the UGV by using the traditional distance-based CL method, proposed configuration optimization CL method, and mainstream CL method are shown in Figure 10.

We can conclude from Figure 10 that the proposed configuration optimization CL method achieves better positioning performance compared to the traditional distance-based CL method. The analysis results based on the RMSE of the UGV’s positioning error are consistent with analysis results based on the UGV’s positioning error curves. Considering that the mainstream CL method has sufficient internal and external source observation constraints, the UGV can achieve better state estimation accuracy and positioning performance. The RMSE of the UGV’s positioning error is 1.79 using the mainstream CL method. Under the condition no assistance from internal sensor information, the proposed configuration optimization CL method has a certain degree of accuracy loss compared to the mainstream CL method. Under the condition of not significantly affecting the positioning performance of the UGV, the number of navigation sensors used for cooperative positioning calculation can be effectively reduced using the proposed method, which to some extent reduces the cost of cluster cooperative positioning.

### 4.2. Experimental Verification and Analysis with Two UAV Nodes

In the CL scenario where there are only two UAVs with known position in the dynamic cluster cooperative network, under the condition of no internal navigation sensor information assistance, due to a lack of sufficient distance observation constraints, the positioning error of the UGV to be located accumulates and diverges over time using the traditional distance-based CL method. In order to verify the performance advantage of the proposed method in this CL scenario, Figure 11 illustrates the positioning error curves of the UGV by using the traditional distance-based CL method, proposed method, and mainstream CL method.

It can be seen from Figure 11 that under the condition of insufficient distance observation constraints in the dynamic cooperative network, the proposed method makes full use of historical distance observation information saved in the navigation processor. Combined with the characteristics of the inertia-based sensor with higher relative position estimation accuracy in a short time period, the position observation equation of the UGV INS is constructed by using distance observation information from historical times and inertia-based relative position constraint information between the UGV and UAVs at adjacent times. The position closed-form solution of the UGV at time $t_{k+1}$ can be obtained in real time using high-precision position information of UAVs from time $t_{k}$ to time $t_{k+1}$, distance observation constraints at adjacent times, and inertia-based relative position constraint from time $t_{k}$ to time $t_{k+1}$. The position closed-form solution of the UGV can be used as the observation vector to estimate the system state variables. Thus, the proposed method achieves better state estimation accuracy and positioning performance compared to the traditional distance-based CL method.

In this scenario, the mainstream CL method makes full use of internal source barometric altimeter information, wheel speed odometer, and external source distance observation information with UAVs as the system observation vectors. The velocity and position observation equations of the UGV INS are constructed using the above-mentioned navigation information sources in the cluster collaborative network. The AEKF is also designed to estimate and compensate system state variables. In order to analyze the CL performance of the proposed method against the mainstream CL method, Table 4 reports the RMSE of the UGV’s positioning error by using the traditional distance-based CL method, proposed method, and mainstream CL method.

Table 4 illustrates that in the case of a dynamic cooperative network with two known-position UAVs and without the assistance of internal source navigation sensor information, the proposed method has a certain degree of accuracy loss compared to the mainstream CL method. The reason is that, due to a lack of sufficient internal source observation constraints, the state estimation accuracy of the UGV using the proposed method cannot be compared to the mainstream CL method. Secondly, the positioning accuracy of the UGV is not only affected by the UAV’s position accuracy and measurement accuracy of the UWB navigation sensor, but the inertia-based relative position constraint error will also be coupled to the position error of the UGV, which reduces the positioning performance of the UGV to a certain extent. However, without the assistance of internal source navigation sensor information, and relying solely on two numbers of relative distance observation constraints to realize the CL calculation, the positioning accuracy of the UGV using the method proposed in this paper can basically meet the practical application requirements in a satellite-denial environment. Under the condition of not significantly affecting the positioning performance of the UGV, the method proposed in this paper can effectively reduce the number of heterogeneous navigation sensors used for collaborative positioning calculation, and reduce the cost of cluster cooperative positioning. In order to more intuitively demonstrate the CL performance of the proposed method in this CL scenario, the positioning error curves of the UGV to be located by using the proposed method and the mainstream CL method are shown in Figure 12.

### 4.3. Experimental Verification and Analysis with Single UAV Node

In the case of a dynamic cluster cooperative network with only one known-position UAV, the traditional distance-based CL method employs a single piece of distance information with the known-position UAV as the observation constraint of the UGV INS; the UGV’s position is located on a spherical surface with the UAV’s position as the center and relative distance as the radius. Indeed, it is impossible for the UGV to gain a three-dimensional spatial position by using a single distance observation constraint. Therefore, the positioning accuracy of the UGV in a satellite-denial environment is difficult to guarantee. The positioning error curves of the UGV to be located by using the traditional distance-based CL method, proposed method, and mainstream CL method are illustrated in Figure 13.

We can conclude from Figure 13 that under the condition of a dynamic cluster cooperative network with single known-position UAV node and without the assistance of any other internal source navigation sensor information, the proposed method can achieve better positioning performance compared to the traditional distance-based CL method. The reason is that distance observation information from historical times received by the UGV are saved in the navigation processor. Combined with high-precision relative position constraints of the inertial sensor in a short time period, the position observation equation of the UGV INS at time $t_{k+2}$ can be constructed using distance observation constraints for three consecutive sampling periods from historical times to the current time and inertia-based relative position constraints from time $t_{k}$ to times $t_{k+1}$ and $t_{k+2}$. Based on the principle of multi-spherical intersection, the method proposed in this paper can obtain a high-precision position closed-form solution for the UGV at time $t_{k+2}$. The position closed-form solution of the UAV at time $t_{k+2}$ can be used as an observation vector to estimate and compensate system state variables. Indeed, the cumulative divergence speed of the UGV’s positioning error is effectively suppressed by using the proposed method.

In this CL scenario, the mainstream CL method employs internal source barometric altimeter information, wheel speed odometer information, and single distance observation constraint information with the UAVs as observation vectors to construct the system velocity and position observation equations. The AEKF is designed to estimate and compensate system state variables. In order to more intuitively illustrate the suppression effect of the proposed method on the UGV’s positioning error, Figure 14 reports the positioning error curves of the UGV by using the proposed method and the mainstream CL method. The RMSE values of the UGV’s positioning error using the traditional distance-based CL method, proposed method, and mainstream CL method are shown in Table 5. It can be seen from Figure 14 that within 210 s cooperative localization time, the positioning error of the UGV to be located is basically maintained within 5 m by using the proposed method and the mainstream CL method. Considering that the inertia-based relative position constraint error at time $t_{k+2}$ is also coupled to the position error of the UGV, compared to the situation where there are two known-position UAVs in a dynamic cooperative network, the positioning performance of the UGV has decreased to a certain degree using the proposed method. However, the positioning accuracy of the UGV to be located can basically meet practical application requirements in a satellite-denial environment. The proposed method fully utilizes distance observation constraints from historical times to construct system observation equations, which can effectively reduce the number of internal source navigation sensors used for CL calculation.

## 5. Conclusions

The traditional distance-based CL method has a certain limitation on the number of known-position UAVs in a cluster cooperative network. As a result of a dynamic change in the number of known-position UAV nodes driven by tasks in the cluster cooperative network, the traditional distance-based CL method becomes unsuitable. Aiming at a cluster cooperative network composed of air-to-ground robots, an adaptive cooperative localization method based on relative distance constraints is proposed in this paper. The adaptive fusion threshold is set based on a dynamic change in the number of UAV nodes in the cluster cooperative network. When there is a large number of known-position UAV nodes in the cluster cooperative network, the GDOP-based configuration optimization strategy is adopted to select the optimal geometric configuration for a cooperative positioning solution. Otherwise, in the case that the number of known-position UAVs is insufficient, the distance observation constraints between the UAV and UGV from historical times are retained in real time. The position observation equation of the UGV INS is constructed by using distance observation constraints from historical times to the current time and inertia-based short-term high-precision relative position constraints. Adaptive extended Kalman filtering is designed to achieve optimal fusion estimation and online compensation of system state variables. The experimental results show that under the condition of a dynamic change in the number of known-position UAV nodes in a cluster cooperative network, the method proposed in this paper can achieve an adaptive CL solution for the UGV, which has better positioning accuracy compared to the traditional distance-based CL method, and reduces the number of internal source navigation sensors used for cooperative localization calculation.

## Figures and Tables

**Figure 1 sensors-24-04543-f001:**
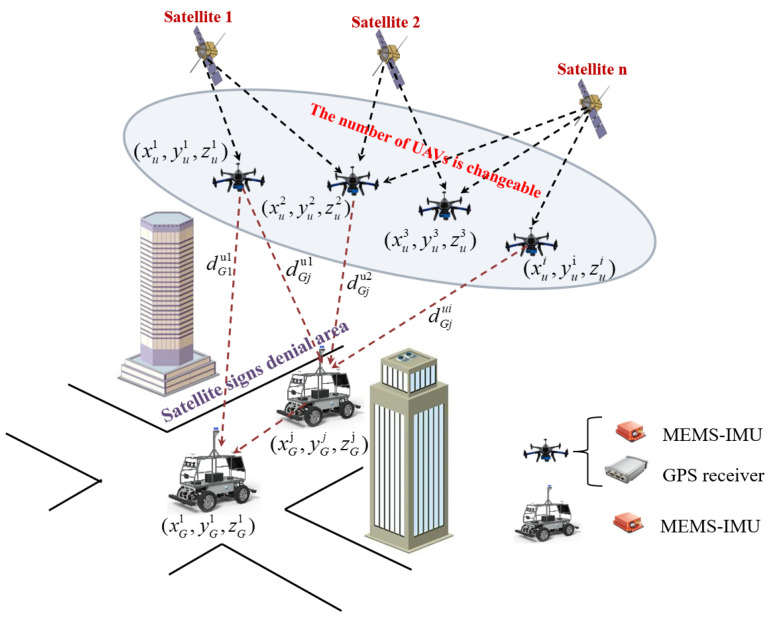
Target-driven cooperative localization scenario for air-to-ground cluster robots based on relative distance constraints.

**Figure 2 sensors-24-04543-f002:**
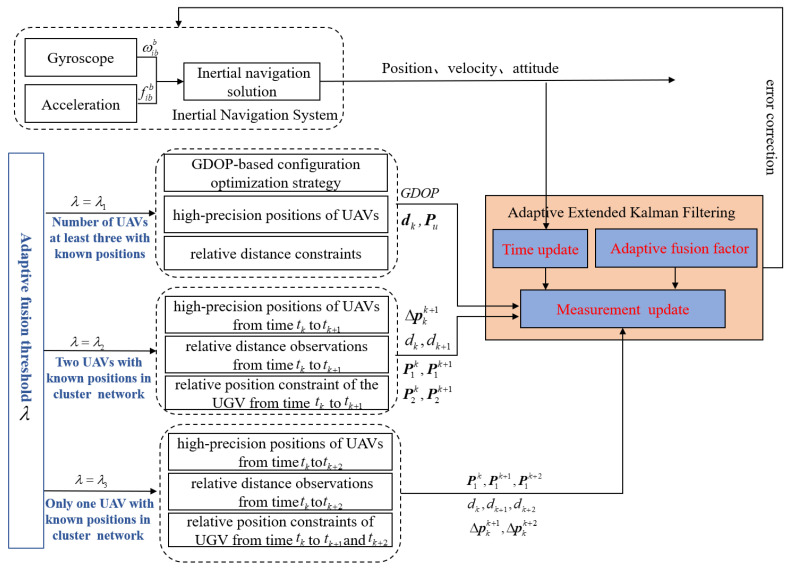
The principle schematic diagram of proposed method in this paper.

**Figure 3 sensors-24-04543-f003:**
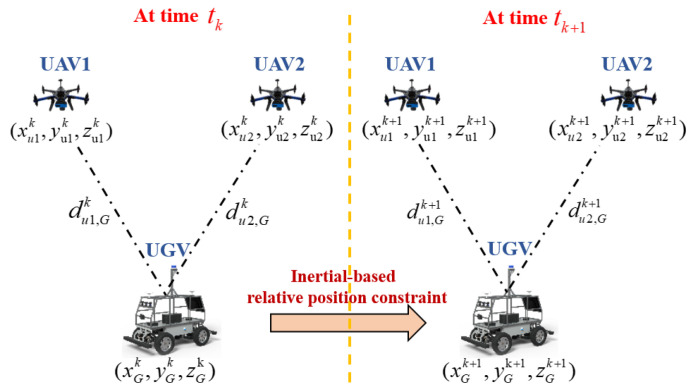
The spatial position relationship between UGV and two known-position UAVs at adjacent times.

**Figure 4 sensors-24-04543-f004:**
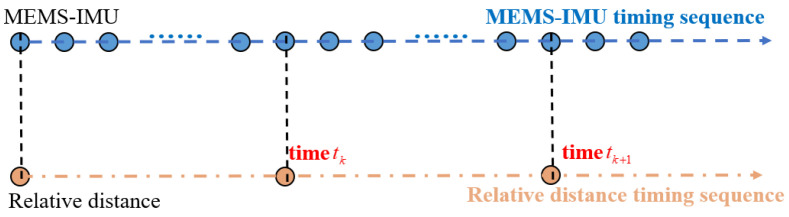
The time sequence schematic diagram of the inertial sensor and relative distance observation between UAV and UGV.

**Figure 5 sensors-24-04543-f005:**
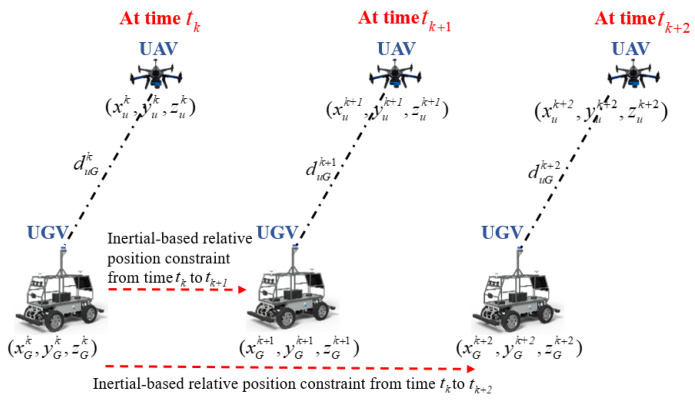
The spatial position relationship between UGV and single known-position UAV at three consecutive relative distance sampling periods.

**Figure 6 sensors-24-04543-f006:**
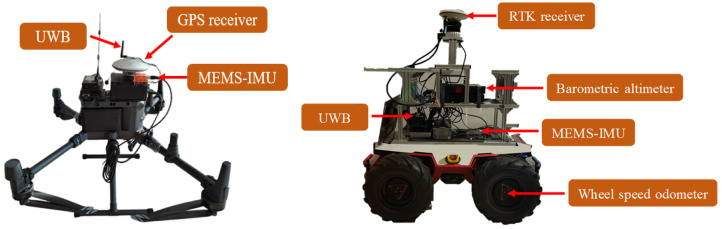
The various navigation sensors mounted on UAV and UGV in the experiment.

**Figure 7 sensors-24-04543-f007:**
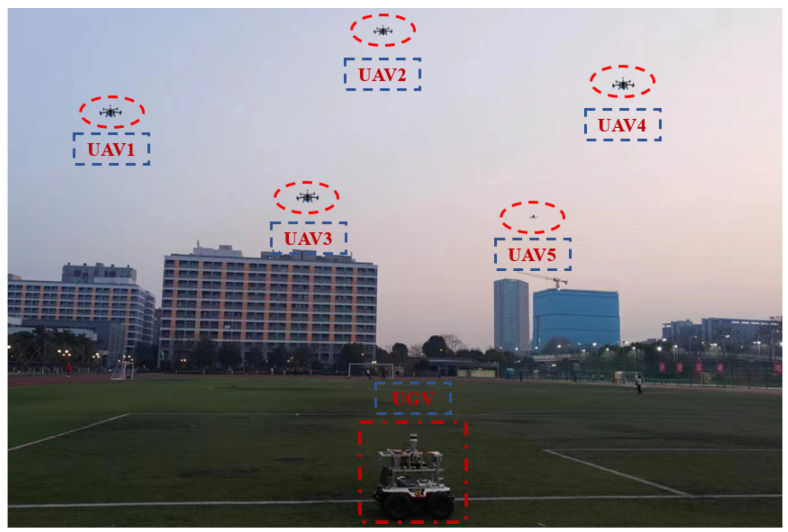
The cooperative localization scenario between UAVs and UGV based on relative distance constraints on outdoor campus playground.

**Figure 8 sensors-24-04543-f008:**
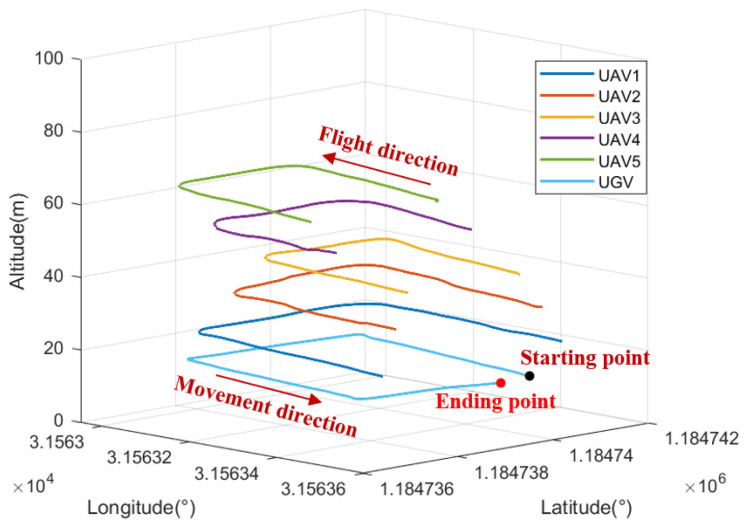
Three-dimensional autonomous motion trajectories of UAV and UGV nodes in experiment.

**Figure 9 sensors-24-04543-f009:**
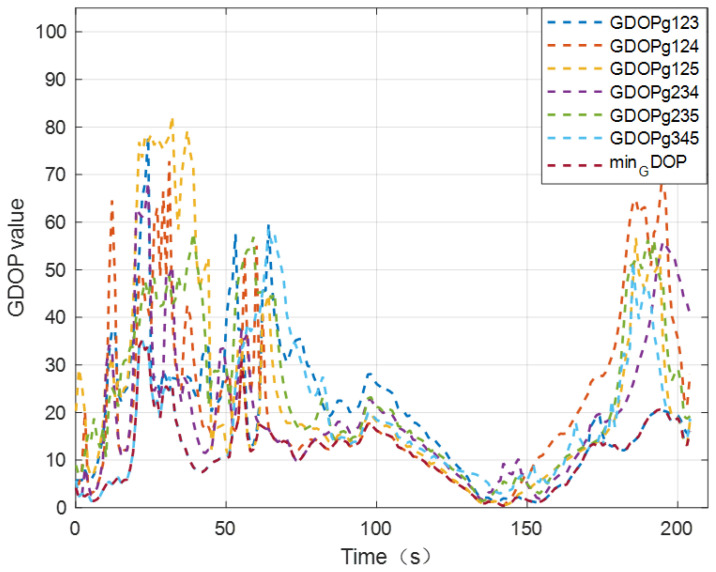
The GDOP value between UGV and UAV nodes in the experiment.

**Figure 10 sensors-24-04543-f010:**
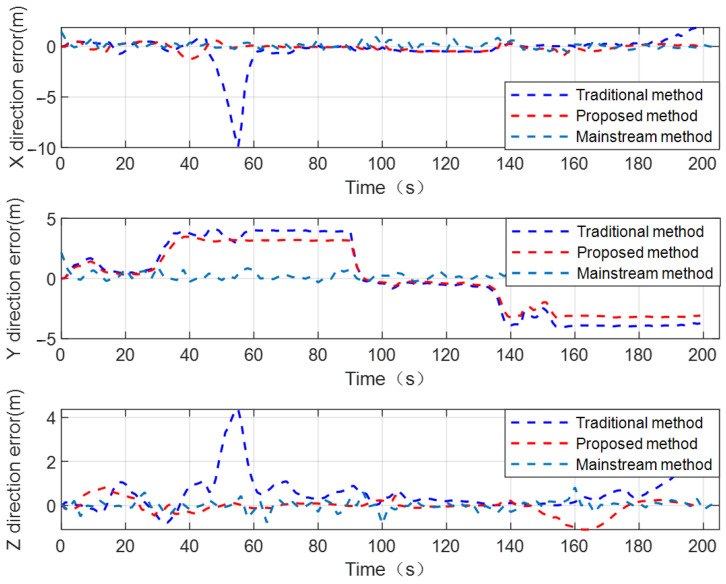
The positioning error curves of UGV by using traditional distance-based CL method, proposed method, and mainstream CL method.

**Figure 11 sensors-24-04543-f011:**
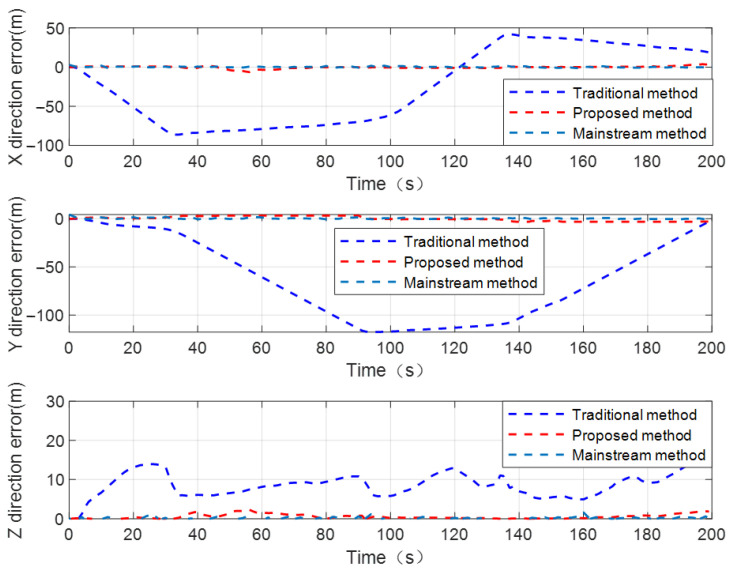
The positioning error curves of UGV using traditional distance-based CL method, proposed method, and mainstream CL method, under dynamic cooperative network with two known-position UAVs.

**Figure 12 sensors-24-04543-f012:**
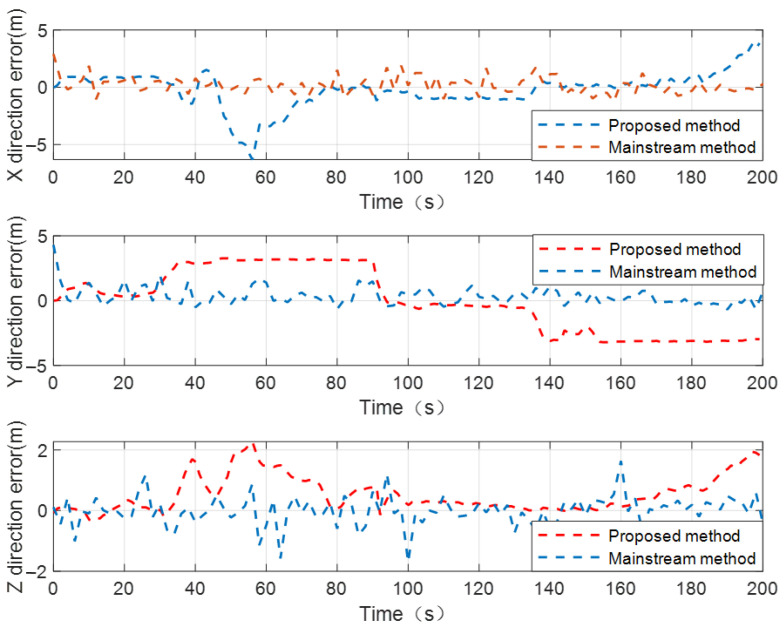
The positioning error curves of UGV to be located by using proposed method and mainstream CL method when there are two known-position UAVs in dynamic cooperative network.

**Figure 13 sensors-24-04543-f013:**
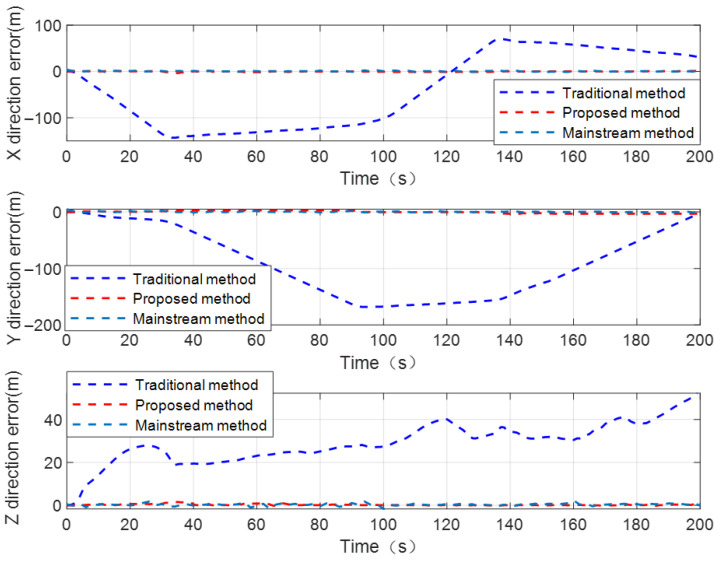
The positioning error curves of UGV to be located by using traditional distance-based CL method, proposed method, and mainstream CL method, under dynamic cooperative network with single known-position UAV.

**Figure 14 sensors-24-04543-f014:**
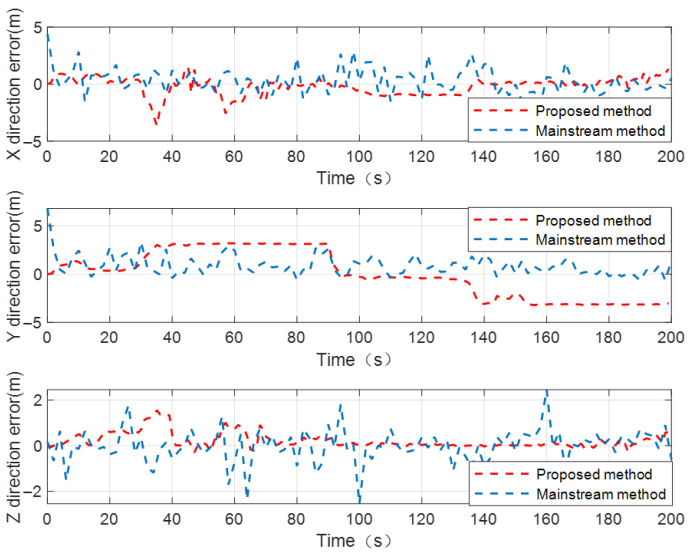
Positioning error curves of UGV to be located by using proposed method and mainstream CL method, under dynamic cooperative network with single known-position UAV.

**Table 1 sensors-24-04543-t001:** The sensor performance parameters used in the experiment.

Sensor Types	Performance Parameters	
Gyroscope of UGV’s MEMS-IMU	Bias stability 15 (°/h)	Random work error 0.2 (${}^{\circ}/\sqrt{h}$)
Accelerometer of UGV’s MEMS-IMU	Bias stability 10 μg	Random work error 10 (μg/$\sqrt{Hz}$)
Gyroscope of UAV’s MEMS-IMU	Bias stability	Random work error
	10 (°/h)	0.5 (${}^{\circ}/\sqrt{h}$)
Accelerometer of UAV’s MEMS-IMU	Bias stability	Random work error
	15 μg	10 (μg/$\sqrt{Hz}$)
UWB/(m)	0.3	
GPS receiver/(m)	[1, 1, 2]	
Barometric altimeter/(m)	0.8	
RTK receiver/(m)	[0.05, 0.05, 0.15]	

**Table 2 sensors-24-04543-t002:** The initial attitude, initial velocity, and initial position information of UAVs and UGV in the experiment.

State Parameters	Parameters Setting
Initial attitude of UAV/(°)	[0, 0, 85]
Initial velocity of UAV/(m/s)	[0, 0, 0]
Initial position of UAV1/(°,°,m)	[118.474124, 31.563485, 21.7437]
Initial position of UAV2/(°,°,m)	[118.474092, 31.563475, 31.7837]
Initial position of UAV3/(°,°,m)	[118.474042, 31.563476, 41.7856]
Initial position of UAV4/(°,°,m)	[118.474021, 31.563400, 52.9768]
Initial position of UAV5/(°,°,m)	[118.473954, 31.563381, 62.1269]
Initial attitude of UGV/(°)	[0, 0, 80]
Initial velocity of UGV/(m/s)	[0, 0, 0]
Initial position of UGV/(°,°,m)	[118.474009, 31.563478, 12.9775]

**Table 3 sensors-24-04543-t003:** Comparison of RMSE value of UGV’s positioning error using distance observation information in various spatial geometric configurations.

GDOP	RMSE.X/(m)	RMSE.Y/(m)	RMSE.Z/(m)	RMSE/(m)
GDOP123	1.50	2.99	0.97	3.48
GDOP124	1.34	2.83	0.86	3.25
GDOP125	1.18	2.78	0.64	3.09
GDOP234	1.03	2.64	0.58	2.89
GDOP235	0.75	2.45	0.51	2.61
GDOP345	0.53	2.41	0.45	2.51
Min GDOP	0.45	2.39	0.42	2.47

**Table 4 sensors-24-04543-t004:** The comparison RMSE of UGV’s positioning error by using traditional distance-based CL method, proposed method, and mainstream CL method.

Method	RMSE.X/(m)	RMSE.Y/(m)	RMSE.Z/(m)	RMSE/(m)
Traditional method	53.75	75.95	9.18	93.49
Proposed method	0.79	2.41	0.41	2.57
Mainstream method	0.69	0.74	0.48	1.12

**Table 5 sensors-24-04543-t005:** The RMSE comparison of UGV’s positioning error using traditional distance-based CL method, proposed method, and mainstream CL method.

Method	RMSE.X/(m)	RMSE.Y/(m)	RMSE.Z/(m)	RMSE/(m)
Traditional method	89.33	108.51	30.45	143.81
Proposed method	1.57	2.42	0.81	2.96
Mainstream method	0.85	0.91	0.59	1.37

## Data Availability

Data are contained within the article.

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
