# Peer review of "An Adaptive Cooperative Localization Method for Heterogeneous Air-to-Ground Robots Based on Relative Distance Constraints in a Satellite-Denial Environment"

_sensors, 2024, doi:10.3390/s24144543_

Round 1

Reviewer 1 Report

Comments and Suggestions for Authors

The paper presents an innovative adaptive Cooperative Localization (CL) method for air-to-ground robots operating in environments without satellite signals. Traditional distance-based CL methods require at least four UAVs with known positions, limiting their applicability when fewer UAVs are available. The proposed method adapts to the dynamic number of UAVs by setting an adaptive fusion estimation threshold. It leverages spatial geometric configurations and relative distance constraints to achieve high-precision localization for Unmanned Ground Vehicles (UGVs). Experimental results demonstrate the effectiveness of the proposed method under varying numbers of known position UAVs.

Comments:

- Clarify the Experimental Setup (Section 4): The description of the experimental setup is somewhat vague, particularly regarding the specific configurations and initial conditions. Provide a detailed schematic or diagram of the experimental setup, including the initial positions, paths, and altitudes of the UAVs and UGV, and the exact placement and specifications of the sensors used.

- Expand on the Theoretical Foundation (Section 3): The theoretical foundation for the adaptive fusion threshold and GDOP-based optimization could be elaborated further. Include a more comprehensive mathematical derivation and explanation of the adaptive fusion threshold and the GDOP optimization strategy, possibly with additional figures or diagrams for clarity.

- Consider studies like ""Modeling of an air quality monitoring network with high space-time resolution 10.1016/B978-0-444-64235-6.50035-8" in the introductory section. In the introduction section of your paper when discussing the importance of high-precision sensor networks in various applications, including environmental monitoring.

- Include More Diverse Test Scenarios (Section 4.1 - 4.3): The experiments are limited to a specific environment and scenario, which may not fully demonstrate the method's robustness. Conduct additional experiments in various environments (e.g., urban, forested) and different weather conditions to validate the method's adaptability and robustness in diverse real-world scenarios.

- Detailed Error Analysis (Section 4.1 - 4.3): The paper provides positioning errors but lacks a detailed error analysis. Include a more detailed error analysis, discussing the sources of errors, their impacts, and how the proposed method mitigates these errors compared to traditional methods.

- Improve the Writing and Structure (Entire Document): Some sections of the paper are difficult to follow due to dense technical language and lack of clear transitions. Enhance the readability by simplifying the language where possible, adding clear transitions between sections, and summarizing key points at the end of each major section. Consider using bullet points or tables to present complex information more clearly.

Reviewer 2 Report

Comments and Suggestions for Authors

This paper proposes an adaptive cooperative localization method for heterogeneous Air-to-Ground Robots Based on relative distance constraints in satellite denial Environment. The main innovation is to adapt to positioning tasks in multiple environments, mainly manifested in adaptive adjustment under the circumstances of redundancy and insufficiency of UAVs at known locations. The proposed algorithm is outstanding in the case of 2 and 1 UAVs, and has strong practicability and innovation. However, the logic of the paper, the completeness of experimental proof and its expression need to be strengthened.

1.There are serious errors in logic and expression. In this paper, the "relative distance constraints" method is derived from the "traditional distance-based" method. However, according to the overview in Introduction, in the multi-robot cooperative positioning algorithm, Both Yang and Qu Y are based on "relative distance constraints", the focus of this paper is whether there is a dynamic adjustment algorithm to accommodate the number of drones in a variable known position. The way this is stated in the abstract is confusing, and it would seem that "relative distance constraints" is the main innovation of the paper. But the main innovation of this paper is "Adaptive Cooperative Localization Method".

2.Therefore, the "Introduction" of this article is also proposed to be modified. First of all, there is no summary of how to obtain the relative distance and the advantages and disadvantages of the method. Second, there is no review of relative navigation algorithms in the absence of UAVs with known locations. Third, if the work in this paper is really based on the "traditional distance-based" method, then it is also necessary to review and modify the relevant content in the abstract.

3.Assuming that the cluster cooperative network is time synchronized by using datalink time precision alignment method, How to ensure that this hypothesis is not explained in the experiment, and how strictly synchronized with the time of UWB and UAV or how much delay is not explained.

4.The text contains 168 lines "equipped with low-precision MEMS-IMU sensors" and 192 lines "using the inertia-based high-precision relative position. "constraints", whether there is a contradiction between low-precision IMU and high-precision relative position constraints, is proposed to be explained in this paper.

5.In this paper, the expression of the adaptive threshold setting is vague. Is it set manually or set by the system itself according to the number of drones in the network? If set artificially, then the innovation and main work of this paper does not seem so clear. If the network is self-regulated, then please give the corresponding expression in the paper, such as how to detect the reliability of the location information of drones in the network and how to determine the fusion threshold based on the number of these drones.

6、If the UWB sensor used in the experiment is reasonable, the paper assumes that UGV is positioned in a satellite-blocked environment, then it is very likely that UGV is in a relatively confined space, while the UAV is in the air, then there must be some obstacles between UGV and UAV, so UWB cannot complete the function of high-precision relative positioning. Please explain in the text.

7.In the experiment, the control experiment of the traditional distance-based CL method is not given the experimental conditions and operation process.

8.For the experiment of a single UAV node, it is recommended to add the experimental control of long-term positioning, because the positioning of the relational navigation instrument in a short time is relatively accurate, so to test its performance, it is recommended to add a performance data after long-term positioning.

Reviewer 3 Report

Comments and Suggestions for Authors

This manuscript presents an adaptive cooperative localization for heterogeneous air-ground robot teams using relative distance constraints in GPS-denied environments.

- In Figure 2, what does "updata" mean? Please explain this structure in more detail in the caption.

- Fig.2 shows the multi-robot solution, but it is hard to see the solution for them ( where is the multi-robot shown? How can we illustrate it?)

- What is the UAV solution to achieve good accuracy? (please discuss and cite)

- The mathematic format style is not good; please rewrite them.

- Please rewrite your contributions to SOTA and the closest works,

- Please compare and analyse the results with those of SOTA. ( not fair to only traditional method)

Comments on the Quality of English Language

N/A

Round 2

Reviewer 2 Report

Comments and Suggestions for Authors

The author has completely answered my question and resolved my doubts.

Reviewer 3 Report

Comments and Suggestions for Authors

Thank you for your efforts,

I have no further comments,

Comments on the Quality of English Language

N/A